# Sleep and Circadian Rhythm in Relation to COVID-19 and COVID-19 Vaccination—National Sleep Survey of South Korea 2022

**DOI:** 10.3390/jcm12041518

**Published:** 2023-02-14

**Authors:** Su-Hyun Han, Seo-Young Lee, Jae Wook Cho, Jee Hyun Kim, Hye-jin Moon, Hea Ree Park, Yong Won Cho

**Affiliations:** 1Department of Neurology, Chung-Ang University Hospital, Chung-Ang University College of Medicine, Seoul 06973, Republic of Korea; 2Department of Neurology, College of Medicine, Kangwon National University, Chuncheon 24341, Republic of Korea; 3Interdisciplinary Graduate Program in Medical Bigdata Convergence, Kangwon National University, Chuncheo 24341, Republic of Korea; 4Department of Neurology, Pusan National University Yangsan Hospital, Pusan National University College of Medicine, Yangsan 50612, Republic of Korea; 5Department of Neurology, Ewha Womans University Seoul Hospital, Ewha Womans University College of Medicine, Seoul 07804, Republic of Korea; 6Department of Neurology, Soonchunhyang University Bucheon Hospital, Bucheon 31151, Republic of Korea; 7Department of Neurology, Inje University College of Medicine, Ilsan Paik Hospital, Goyang 10380, Republic of Korea; 8Department of Neurology, Keimyung University School of Medicine, 1095 Dalgubeol-daero, Dalseo-gu, Daegu 42601, Republic of Korea

**Keywords:** sleep, circadian rhythm, chronotype, COVID-19, vaccination

## Abstract

Background: Currently, information on sleep and circadian patterns in relation to COVID-19 or vaccination remains limited. We aimed to investigate sleep and circadian patterns according to history of COVID-19 and COVID-19 vaccination side effects. Methods: We used data from the National Sleep Survey of South Korea 2022, a nationwide cross-sectional population-based survey regarding sleep–wake behaviors and sleep problems among Korean adults. Analysis of covariance (ANCOVA) and logistic regression analyses were performed to explore the different sleep and circadian patterns according to the history of COVID-19 or self-reported side effects of the COVID-19 vaccination. Results: The ANCOVA showed that individuals with a history of COVID-19 presented a later chronotype than individuals without a history of COVID-19. Individuals who had experienced vaccine-related side effects had a shorter sleep duration, poorer sleep efficiency, and worse insomnia severity. Multivariable logistic regression analysis showed a later chronotype related to COVID-19. A short sleep duration, poorer sleep efficiency, and worse insomnia severity were associated with self-reported side effects of the COVID-19 vaccination. Conclusions: Individuals who recovered from COVID-19 had a later chronotype than those without a history of COVID-19. Individuals who had experienced vaccine-related side effects presented with poorer sleep than those without side effects.

## 1. Introduction

It has been approximately three years since the global outbreak of the severe acute respiratory syndrome coronavirus 2 (SARS-CoV-2) virus and the sequential coronavirus disease 2019 (COVID-19). The pandemic, lockdowns, social restrictions, fear of the illness, and uncertainty have changed our lives and brought psychological problems [1]. A recent meta-analysis showed a high prevalence of sleep problems [2], and sleep has been one of the top concerns during this pandemic period.

A growing body of research indicates that sleep disturbances can lead to serious consequences on individuals’ health and quality of life [3]. Chronotypes might also impact general health, including physical and psychological outcomes [4]. A chronotype refers to an individual’s preferred timing for sleep and activity [4]. Morningness (individuals colloquially called “larks”) corresponds to an earlier sleep–wake schedule than eveningness (individuals colloquially called “owls”). Individuals with different chronotypes present significant variations in psychological, behavioral, and biological aspects [5]. Eveningness may be associated with an increased risk of metabolic disorders and mental health problems including depressive and anxiety symptoms, behavioral dysregulation, and unhealthy behavior [4]. Eveningness is also related to the stress personality traits of an individual [6].

Patients with active COVID-19 have been shown to have a higher prevalence (74.8%) of sleep disturbances than the rest of the population (35.7%) [2]. Even after recovery from COVID-19, neuropsychiatric sequelae such as sleep disturbances are increasingly being reported [7,8]. Although COVID-19 is primarily considered a respiratory disease, it can affect multiple organ systems, including the central nervous system [9]. It remains to be explored whether COVID-19 has long-term effects on the circadian rhythm and sleep. In addition, sleep and circadian rhythms play an important role in immunological functions [10]. Recent studies have suggested that sleep problems may be a risk factor for COVID-19 [11]. However, limited data exist on the association of sleep and circadian rhythms with COVID-19, and further research is needed.

There have been extensive efforts to overcome the COVID-19 pandemic, and vaccines have been widely distributed and administered to large populations [12,13]. However, vaccine hesitancy still exists [12,13]. Safety concerns and side effects of the vaccines, including fever, headache, fatigue, chills, and pain at the injection site, represent significant reasons for the vaccine hesitancy [12,13]. In addition to identifying direct side effects, it is also necessary to investigate the psychological consequences, such as sleep problems, in patients who experience various vaccine-related side effects to strategize against the vaccine hesitancy. Furthermore, psychological factors may also be implicated in the prevalence and severity of vaccine-related adverse effects [14]. However, few studies have been conducted to evaluate whether sleep problems are related to these adverse events. The association of sleep with vaccine efficacy is well known, and a good night’s sleep can positively influence the immune response resulting in higher vaccination efficacy [14,15,16]. Investigations on sleep and the occurrence of various vaccine-related side effects, regardless of the direction of causality, may suggest the importance of sleep management before and after vaccination and may help to promote safe vaccination.

Therefore, this study aimed to investigate sleep and circadian patterns according to the history of COVID-19 or the experience of side effects from the COVID-19 vaccination.

## 2. Materials and Methods

### 2.1. Subjects and Survey Procedure

This study used data from the National Sleep Survey of South Korea 2022 (NSSK), a nationwide population-based survey of sleep disturbances among Korean adults 21 to 69 years old. An online survey using a structured questionnaire was conducted from January 2022 to February 2022. A representative sample of 4000 participants was constituted using a stratified, multistage random sampling method based on sex, age, and place of residence. This survey was conducted by the Embrain Public company on behalf of the Epidemiology Committee of the Korean Sleep Research Society. This study was approved by the Institutional Review Board of the Dongsan Medical Center (IRB No. 2021-12-063).

### 2.2. Questionnaires

A structured web questionnaire consisting of questions about demographics, health habits, current sleep patterns, history of COVID-19 and COVID-19 vaccination, modified Insomnia Severity Index (ISI), and Epworth Sleepiness Scale (ESS) was utilized.

All participants were asked if they had ever had COVID-19 and whether they had been vaccinated. Participants who received two or more doses of the COVID-19 vaccine were asked if they had any adverse reactions to the vaccine. We used the following questions: “Have you ever had COVID-19?”, “Have you received two or more doses of the COVID-19 vaccine?”, and “Have you experienced any side effects from the COVID-19 vaccination?” (Figure 1).

We investigated usual sleep times on workdays and free days. The average sleep time was calculated as follows: (weekday sleep time × 5 + weekend sleep time × 2)/7. Sleep efficiency was defined as the ratio of average sleep duration to average time in bed.

To measure chronotype, we used mid-sleep on free days corrected for sleep debt on workdays (MSFsc), using the formula: MSFsc = midsleep point on free days − (sleep duration on free days − average weekly sleep duration)/2. A higher MSFsc value reflects a stronger eveningness tendency. Based on the MSFsc distribution of their sample, with 2.5% at each end of the distribution as the extreme chronotypes, Kühnle et al. [17] suggested that an MSFsc value less than 2.17 should be defined as extreme morningness and that an MSFsc value of 7.25 or greater should be defined as extreme eveningness. We used the midsleep point on free days (MSF) and the midsleep point on workdays (MSW) to quantify social jet lag using the formula: Social jetlag = |MSF − MSW|. The definition and measurement refer to the theory and process presented by Wittmann et al. [18].

We used the Korean version of the ISI to investigate insomnia symptoms [19]. The optimal cut-off score of the ISI was 15.5, and the sensitivity and specificity at that score were 0.92 and 0.82, respectively [19]. The reliability was confirmed by Cronbach’s alpha of 0.92, and the item-to-total-score correlations (item–total correlations) ranged from 0.65 to 0.84 [19]. The ISI is a brief screening questionnaire that measures insomnia severity. The total score ranges from 0 to 28, with a higher score indicating greater insomnia severity. If a participant’s ISI score was 15 or more, they were classified as having moderate to severe insomnia. In this study, difficulty initiating sleep, difficulty maintaining sleep, and waking up too early were evaluated using the criteria that these issues were experienced five or more times per week.

The Korean version of the ESS is a reliable and valid instrument for screening patients with daytime sleepiness [20]. The ESS consists of eight sleep-related situations, and participants were asked to assess the likelihood of falling asleep in each situation. The total score ranges from 0 to 24, with a higher score indicating a high level of daytime sleepiness. An ESS score of 11 or more was classified as excessive daytime sleepiness.

### 2.3. Statistical Analysis

Data are described as the mean ± standard deviation (*SD*) or number (%). We compared sleep and circadian rhythms according to the history of COVID-19 or the self-reported side effects of the COVID-19 vaccination. An analysis of covariance (ANCOVA) adjusted for age, sex, residential area, marital status, employment, education level, monthly income level, body mass index (BMI), alcohol consumption, smoking, and coffee consumption was used.

Logistic regression analyses were performed to explore the association between sleep and circadian patterns with COVID-19 or the self-reported side effects of the COVID-19 vaccination. The average sleep latency, average sleep duration, social jet lag, sleep efficiency, MSFsc, ESS, and ISI were each used as independent variables. To analyze the independent effect of each sleep parameter, a multivariable analysis was performed after adjusting for age, sex, BMI, alcohol consumption, and smoking.

Statistical analyses were performed using the Statistical Package for the Social Sciences (SPSS, version 19.0, Chicago, IL, USA). For all analyses, the significance threshold was set at a *p*-value of less than 0.05.

## 3. Results

### 3.1. Baseline Characteristics of the Total Population

Table 1 presents the sociodemographic characteristics of the 4000 participants. Forty-eight individuals had a history of COVID-19, and 3704 individuals had received two or more COVID-19 vaccines. Among them, 1529 individuals had experienced side effects of the COVID-19 vaccination. Table 2 shows the sleep patterns and circadian rhythms of the participants. Of the 4000 participants, 516 (12.9%) individuals were classified as having moderate to severe insomnia (Table 2). Difficulty initiating sleep, difficulty maintaining sleep, or waking up too early more than five times per week was reported in 3.6, 6.8, and 7.4% of the total participants, respectively (Table 2). The sleep efficiency was 88.8% (*SD* = 12.2) (Table 2).

### 3.2. COVID-19 History

In Table 3, sleep patterns and circadian rhythms according to the history of COVID-19 are presented after adjusting for sociodemographic variables such as age, sex, residential area, marital status, employment, education level, monthly income level, BMI, alcohol consumption, smoking, and coffee consumption. Individuals with a history of COVID-19 (*n* = 48) presented a later chronotype than individuals without a history of COVID-19 (*p* < 0.01) (Table 3).

Logistic regression analyses were performed to explore the association of sleep and circadian patterns with COVID-19 (Table 4). A later chronotype was associated with a history of COVID-19 (*p* < 0.01), and the association was still significant after adjusting for age, sex, BMI, smoking, and alcohol consumption (*p* = 0.002) (Table 4).

### 3.3. Self-Reported Vaccine-Related Adverse Effects

Among individuals who had received two or more COVID-19 vaccinations, the sleep patterns and circadian rhythms according to the self-reported vaccine-related adverse effects are presented in Table 5. The values are adjusted for sociodemographic variables such as age, sex, residential area, marital status, employment, education level, monthly income level, BMI, alcohol consumption, smoking, and coffee consumption. Individuals with self-reported vaccination side effects (*n* = 1529) showed a shorter sleep duration (*p* = 0.04), poorer sleep efficiency (*p* = 0.01), and worse insomnia severity (*p* < 0.01) than individuals without vaccine-related side effects (Table 5).

Logistic regression analyses were performed to explore the association of sleep and circadian patterns with the self-reported side effects of the COVID-19 vaccination (Table 6). In individuals who had received two or more COVID-19 vaccinations, longer sleep latency (*p* < 0.001), poorer sleep efficiency (*p* = 0.001), longer social jet lag (*p* = 0.044), later chronotype (*p* < 0.001), and worse insomnia severity (*p* < 0.001) were associated with the self-reported side effects of the COVID-19 vaccination. Multivariable analyses were performed after adjusting for the sociodemographic variables, and shorter sleep duration (*p* = 0.03), poorer sleep efficiency (*p* = 0.01), and worse insomnia severity (*p* < 0.01) were associated with the self-reported side effects of the COVID-19 vaccination. 

## 4. Discussion

Sleep and circadian rhythms play an important role in immunological functions [10]. A recent study identified that circadian misalignment, also known as social jet lag, is associated with COVID-19 [21]. Henríquez-Beltrán et al. also reported circadian disruption, including significant fragmentation of the rest–activity rhythm, four months after COVID-19 [22]. However, investigations of the chronotype after recovery from COVID-19 are scarce. In our investigation, we identified that individuals who recovered from COVID-19 had a later chronotype than those without a history of COVID-19. A previous study reported that 23.4% of SARS-CoV-2-positive participants shifted to an earlier bedtime, nearly double that observed among the SARS-CoV-2-negative (12.2%) and untested (10.2%) participants [23]. However, their finding may suggest that acute infectious illnesses induce a sickness behavior characterized by fatigue and sleepiness [24]. To date, few studies have been conducted on the chronotype as a long-term consequence after recovery from COVID-19, and whether COVID-19 perturbs the circadian rhythm after recovery remains to be explored. Song et al. used human brain organoids and mice, which overexpressed human angiotensin-converting enzyme-2, to demonstrate that SARS-CoV-2 can infect neurons [9]. Furthermore, SARS-CoV-2 was detected in the cortical neurons of the brain during the autopsies of patients with COVID-19 [9]. It is tempting to speculate that the SARS-CoV-2 infection of neurons may lead to the perturbation of the circadian rhythms in patients with COVID-19 [16].

Abdelghani et al. reported that individuals who had recovered from COVID-19 were more likely to have prolonged sleep latency, shorter sleep duration, and reduced sleep efficiency [7]. In contrast, Donezella et al. reported that laboratory-confirmed SARS-CoV-2-positive participants reported a longer sleep duration than SARS-CoV-2-negative participants more than 30 days after infection [23]. Although the results are different and direct comparison is difficult because of the different methodologies, these studies commonly suggest that alterations in sleep patterns can follow COVID-19. Recent studies have suggested sleep difficulties as a potential sequela of COVID-19 [7,23]. Individuals with COVID-19 were found to have a higher probability of having trouble sleeping than individuals without COVID-19 after more than 30 days [23]. Patients with COVID-19 still experienced sleep difficulties six months after recovery [25]. In a UK primary care registry, there was consistent evidence that SARS-CoV-2 infections were associated with increased risks of sleep problems more than six months after the infection [26]. In addition, they reported that a longer follow-up period was associated with a greater risk of sleep problems [26]. In our study, there were no differences in sleep latency, duration, and efficiency. Individuals with a history of COVID-19 tended to report more daytime sleepiness (*p* = 0.08) and insomnia (*p* = 0.09) than individuals without a history of COVID-19, but this difference was not statistically significant. The lack of statistical significance in this study may be due to the low prevalence of COVID-19 participants.

Another important aspect of the pandemic was vaccination issues. There have been few documented direct side effects related to sleep associated with the COVID-19 vaccine. Some studies have suggested hypersomnia after the COVID-19 vaccination [27,28]. Additionally, some people have complained of decreased sleep quality after receiving a COVID-19 vaccination [29,30]. To our knowledge, sleep consequences as a vaccine-related side effect have not been investigated. In our investigation, participants who had experienced vaccine-related side effects had a shorter sleep duration, poorer sleep efficiency, and worse insomnia severity than those who did not have adverse effects. There were no differences in social jet lag, chronotype, or daytime sleepiness between participants with and without self-reported side effects. We identified that individuals might suffer from sleep disturbances after experiencing vaccine-related side effects. Given that sleep deprivation is a significant effect of vaccine efficacy [14,15,16], our results suggest the importance of sleep management before and after COVID-19 vaccination. Sleep management is important to promote safe COVID-19 vaccination and reduce vaccine hesitancy.

The study has some limitations. First, although their characteristics were quite similar to the source population, the participants were included on a voluntary basis. Second, although we focused on sleep and circadian patterns according to the history of COVID-19 or the experience of vaccine-related side effects, the design was cross-sectional, and thus, causal relationships cannot be determined. A large-scale prospective study is necessary to confirm our findings. Third, we used a self-reported questionnaire about the history of COVID-19 and the experience of vaccine-related side effects. We did not collect information on the type of side effects experienced. We also did not determine the elapsed time since recovery from COVID-19. Fourth, we did not investigate psychiatric symptoms, which might be significant contributors to sleep problems in the context of the SARS-CoV-2 pandemic. Finally, at the time of the survey, the prevalence of COVID-19 was very low (1.2%) in our population, causing an imbalance in the numbers between the groups. At the midpoint of this survey (31 January 2022), the cumulative number of patients was estimated to be approximately 8.45 million [31] (approximately 1.6% of the total population of Korea), but the current cumulative number of patients has exceeded approximately 30 million (3 February 2023). Further research is needed to more accurately evaluate the current situation in which the infection is much more widespread.

## 5. Conclusions

We identified that individuals who recovered from COVID-19 had a later chronotype than those without a history of COVID-19. However, there were no differences in other sleep patterns. In addition, individuals who had experienced side effects from the COVID-19 vaccine presented with poorer sleep than those who did not experience side effects. Sleep and circadian patterns are essential to our lives and are important aspects during the SARS-CoV-2 pandemic. The SARS-CoV-2 pandemic is still ongoing, and another viral pandemic may occur. Further studies should research how sleep and circadian rhythms are related to COVID-19 or the COVID-19 vaccination. Sleep and circadian rhythms should be considered during strategic planning to combat the current and future pandemics.

## Figures and Tables

**Figure 1 jcm-12-01518-f001:**
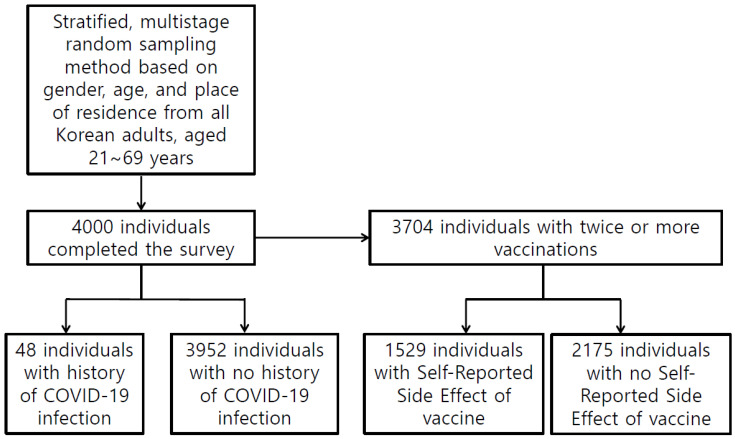
Study Flow.

**Table 1 jcm-12-01518-t001:** Sociodemographic characteristics in the total participants.

Variables	Category	Total Participants (*N* = 4000)
Age, year		44.8 ± 13.3
Sex	Male	2035 (50.9%)
	Female	1965 (49.1%)
Residential area	Urban	1789 (44.7%)
	Rural	2211 (55.3%)
Marital status	Married	2338 (58.5%)
	Unmarried	1437 (35.9%)
	Divorced/bereaved	225 (5.6%)
Employment	Employed	2852 (71.3%)
	Unemployed	1148 (28.7%)
Education level	Middle school or less	35 (0.9%)
	High school	920 (23.0%)
	College or more	3045 (76.1%)
Monthly income level	<2,000,000	517 (12.9%)
	2,000,000–4,999,999	1974 (49.4%)
	≥5,000,000	1509 (37.7%)
Body mass index, kg/m^2^		23.7 ± 13.5
Alcohol	No	1306 (32.7%)
	Yes	2694 (67.4%)
Smoking	No	1862 (46.6%)
	Yes	2138 (53.5%)
Coffee	<1/day	1331 (33.3%)
	≥1/day	2669 (66.7%)

Data are described as the mean ± standard deviation or number (%).

**Table 2 jcm-12-01518-t002:** Sleep patterns and circadian rhythms of the total participants.

Variables	Total Participants (*N* = 4000)
Sleep latency_weekday, min	30.6 ± 26.6
Sleep latency_weekend, min	34.2 ± 29.1
Average sleep latency, min	31.7 ± 26.4
Sleep duration_weekday, hour	6.6 ± 1.3
Sleep duration_weekend, hour	7.6 ± 1.5
Average sleep duration, hour	6.9 ± 1.2
Social jet lag, min	64.7 ± 67.3
Sleep efficiency, %	88.8 ± 12.2
Chronotype, MSFsc	4:11 ± 2:16
Epworth Sleepiness Scale	6.5 ± 3.6
Insomnia Severity Index	9.4 ± 4.6
Moderate to severe insomnia	516 (12.9%)
Difficulty initiating sleep	145 (3.6%)
Difficulty maintaining sleep	273 (6.8%)
Waking up too early	294 (7.4%)

Data are described as the mean ± standard deviation or number (%). MSFsc, mid-sleep time on free days corrected for sleep debt accumulated over the work week.

**Table 3 jcm-12-01518-t003:** Sleep and circadian rhythm according to COVID-19 infection history.

Variables	COVID-19 Infection (+)	COVID-19 Infection (−)	*p*
	*N* = 48	*N* = 3952	
Average sleep latency, min	34.71 ± 3.70	31.64 ± 0.41	0.41
Average sleep duration, hour	6.80 ± 0.18	6.91 ± 0.02	0.52
Social jet lag, min	75.52 ± 9.41	64.59 ± 1.03	0.25
Sleep efficiency, %	86.09 ± 1.76	88.86 ± 0.19	0.12
Chronotype, MSFsc	05:11 ± 00:18	04:10 ± 00:02	<0.01
Epworth Sleepiness Scale	7.32 ± 0.51	6.50 ± 0.06	0.11
Insomnia Severity Index	10.54 ± 0.67	9.38 ± 0.07	0.09

Analysis of covariance was performed, and the table shows the mean ± standard error adjusted for age, sex, residential area, marital status, employment, education level, monthly income level, body mass index, alcohol consumption, smoking, and coffee consumption. MSFsc, mid-sleep time on free days corrected for sleep debt accumulated over the work week.

**Table 4 jcm-12-01518-t004:** Association of sleep pattern with COVID-19 infection.

	COVID-19 Infection			
	β	SE	*p*-Value	OR (95% CI)
Unadusted				
Average sleep latency	<0.01	0.01	0.44	1.00 (0.99–1.01)
Average sleep duration	−0.07	0.12	0.54	0.93 (0.74–1.17)
Sleep efficiency	−0.01	0.01	0.14	0.99 (0.97–1.00)
Social jet lag	<0.01	<0.01	0.10	1.00 (1.00–1.00)
Chronotype, MSFsc	−0.14	0.04	<0.01	1.15 (1.06–1.24)
Epworth Sleepiness Scale	0.07	0.04	0.08	1.07 (0.99–1.15)
Insomnia severity index	0.05	0.03	0.07	1.05 (1.00–1.11)
Adjusted ^†^				
Average sleep latency	<0.01	0.01	0.63	1.00 (0.99–1.01)
Average sleep duration	−0.09	0.12	0.43	0.91 (0.73–1.14)
Sleep efficiency	−0.01	0.01	0.15	1.00 (0.97–1.01)
Social jet lag	<0.01	<0.01	0.14	1.00 (1.00–1.00)
Chronotype, MSFsc	0.13	0.04	<0.01	1.13 (1.05–1.23)
Epworth Sleepiness Scale	0.07	0.04	0.08	1.07 (0.99–1.15)
Insomnia Severity Index	0.05	0.03	0.09	1.05 (0.99–1.11)

β, regression coefficient; SE, standard error; OR, odds ratio; CI, confidence interval. ^†^ Adjusted for sociodemographics (age, gender, body mass index, smoking, and alcohol consumption).

**Table 5 jcm-12-01518-t005:** Sleep and circadian rhythms according to self-reported side effects of COVID-19 vaccination.

Variables	Self-Reported Side Effects of Vaccination (+)	Self-Reported Side Effects of Vaccination (−)	*p*
	*N* = 1529	*N* = 2175	
Average sleep latency, min	32.39 ± 0.66	30.97 ± 0.55	0.11
Average sleep duration, hour	6.85 ± 0.03	6.93 ± 0.03	0.04
Social jet lag, min	65.15 ± 1.62	64.50 ± 1.35	0.76
Sleep efficiency, %	88.27 ± 0.31	89.31 ± 0.26	0.01
Chronotype, MSFsc	04:11 ± 00:03	04:08 ± 00:03	0.49
Epworth Sleepiness Scale	6.66 ± 0.09	6.45 ± 0.08	0.08
Insomnia Severity Index	9.93 ± 0.12	9.03 ± 0.10	<0.01

Analysis of covariance was performed, and the table shows the mean ± standard error adjusted for age, sex, residential area, marital status, employment, education level, monthly income level, body mass index, alcohol consumption, smoking, and coffee consumption. MSFsc, mid-sleep time on free days corrected for sleep debt accumulated over the work week.

**Table 6 jcm-12-01518-t006:** Association of sleep pattern with self-reported side effects of COVID-19 vaccination.

	Self-Reported Side Effects of Vaccination			
	β	SE	*p*-Value	OR (95% CI)
Unadjusted				
Average sleep latency	0.01	<0.01	<0.01	1.01 (1.00–1.01)
Average sleep duration	−0.01	0.03	0.74	0.99 (0.94–1.05)
Sleep efficiency	−0.01	<0.01	<0.01	0.99 (0.99–1.00)
Social jet lag	<0.01	<0.01	0.04	1.00 (1.00–1.00)
Chronotype, MSFsc	0.06	0.02	<0.01	1.06 (1.03–1.10)
Epworth Sleepiness Scale	0.02	0.01	0.08	1.02 (1.00–1.04)
Insomnia severity index	0.05	0.01	<0.01	1.05 (1.03–1.06)
Adjusted ^†^				
Average sleep latency	<0.01	<0.01	0.14	1.00 (1.00–1.01)
Average sleep duration	−0.06	0.03	0.03	0.94 (0.89–1.00)
Sleep efficiency	−0.01	<0.01	0.01	0.99 (0.99–1.00)
Social jet lag	<0.01	<0.01	0.66	1.00 (1.00–1.00)
Chronotype, MSFsc	0.03	0.02	0.12	1.03 (0.99–1.06)
Epworth Sleepiness Scale	0.02	0.01	0.07	1.02 (1.00–1.04)
Insomnia Severity Index	0.04	0.01	<0.01	1.04 (1.03–1.06)

β, regression coefficient; SE, standard error; OR, odds ratio; CI, confidence interval. † Adjusted for sociodemographics (age, gender, body mass index, smoking, and alcohol consumption).

## Data Availability

Data are available upon reasonable request to the corresponding author.

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
