# Peer review of "Sleep and Circadian Rhythm in Relation to COVID-19 and COVID-19 Vaccination—National Sleep Survey of South Korea 2022"

_jcm, 2023, doi:10.3390/jcm12041518_

Round 1
Reviewer 1 Report
It's well-written research, with interesting topics and new results. I congratulate the Authors for the work done. However, I do have some suggestions for improvement in this manuscript.
1. Introduction
Written concisely, however, it lacks introductory information on the subject of sleep and circadian rhythm and the factors associated with them, and the effects of their disruption.
It is important that individual preferences as for falling asleep, waking up, increased mental and physical activity have been the basis for distinguishing various types of circadian activity, i.e. the so-called chronotypes. The types are: M (Morning, "larks") and E (Evening, "owls"). The manifestation of a given chronotype depends on a number parameters: genetic predispositions, age, sex, socioeconomic factors, exposure to light, season of birth. The academic studies carried out so far have indicated a link between a given chronotype (M or E) and the sleep cycle and effectiveness, lifestyle regularity, level of anxiety, mood changes. The evening chronotype has been associated with an increased risk for mood disorders, such as major depressive disorder (MDD), bipolar disorder and seasonal affective disorder (SAD). Increasingly, evening chronotype has been linked with mental health problems beyond mood disorders such as attention deficits, anxiety, alcohol dependence, and antisocial behaviors, suggesting that evening chronotype may constitute a transdiagnostic risk factor more broadly.
Circadian activity is also related to the personality traits of the individual, which is worth mentioning in the manuscript[Chronotype Profile, Stress, Depression Level, and Temporomandibular Symptoms in Students with Type D Personality. Journal of Clinical Medicine. 2022; 11(7):1886. https://doi.org/10.3390/jcm11071886].
.
2. Methods
a) The description of the scale used should include the range of scores, cut-offs, validity, and reliability.
b) It is worth adding an attachment (supplement) with the questionnaires used.
3. Results
In tables 3-6, it is enough to enter values up to two decimal places.
Reviewer 2 Report
Thank you for the opportunity to review this interesting manuscript. This is a survey-based study of sleep patterns in relation to self-reported COVID vaccination status among a presumably representative sample of South Korean adults.
The major weakness of this analysis is its reliance on self-reported data, certain of which seems invalid at face value.
In particular, it is not clear to me how or why a presumably representative sample of 4,000 South Korean adults, surveyed in 2022, would report such a relatively low prevalence of past COVID infection.
According to current statistics from Statistics Korea, as of the date of this review in early 2023, there have been at least 30 million cumulative COVID infections, among a population of over 50 million, implying an approximate prevalence of 60%.
Even assuming a slightly lower prevalence at the time of this survey’s completion in 2022, this study’s reported prevalence of past infection of over 1% differs so significantly from what would be expected, that the validity of other self-reported data must immediately be called into question.
Accordingly, as the authors concede in their introduction that past COVID infection can result in “neuropsychiatric sequelae such as sleep disturbances”, the validity of the authors’ attribution of such observations in the current study to COVID vaccination status, and not prior infection, cannot be validly made.
The authors are encouraged to provide an explanation for the apparent discrepancy in COVID prevalence reported here, and to comment on the potential role of reporting errors on the validity of their conclusions, including specifically those arising from the analysis in Tables 3 and 4, but also from those in Tables 5 and 6, under an assumption that some subjects may have misreported their COVID infection history.
Round 2
Reviewer 2 Report
I thank the authors for their helpful comments in response to my original concerns. The work is significant improved as a result of their comments and clarifications, particularly in regards to the timing of the COVID epidemic in relation to the period of survey administration.
I have no concerns with the other minor changes made to the manuscript by the authors in response to this opportunity, which appear to be slight clarifications and changes in formatting.
The work is, overall, significantly improved. I have no further concerns.